# Quantifying Cross-Modality Memorization in Vision-Language Models

**Yuxin Wen**[1]*, **Yangsibo Huang**[2], **Tom Goldstein**[1]
**Ravi Kumar**[2], **Badih Ghazi**[2], **Chiyuan Zhang**[2]
[1]University of Maryland, College Park
[2]Google

## Abstract

Understanding what and how neural networks memorize during training is crucial, both from the perspective of unintentional memorization of potentially sensitive information and from the standpoint of effective knowledge acquisition for real-world, knowledge-intensive tasks. While previous studies primarily investigate memorization within a single modality, such as text memorization in large language models or image memorization in diffusion models, unified multimodal models are becoming increasingly prevalent in practical applications. In this work, we focus on the unique characteristics of *cross-modality* memorization and conduct a systematic study centered on vision-language models. To facilitate controlled experiments, we first introduce a synthetic persona dataset comprising diverse synthetic person images and textual descriptions. We quantify factual knowledge memorization and cross-modal transferability by training models on a single modality and evaluating their performance in the other. Our results reveal that facts learned in one modality transfer to the other, but a significant gap exists between recalling information in the "source" and "target" modalities. Furthermore, we observe that this gap exists across various scenarios, including more capable models, machine unlearning, and the multi-hop case. At the end, we propose a baseline method to mitigate this challenge. We hope our study can inspire future research on developing more robust multimodal learning techniques to enhance cross-modal transferability.

## 1 Introduction

Modern foundation models are continuing to benefit from scaling with more training data. Large volumes of diverse, high-quality data allow the models to develop fundamental capabilities like language understanding and reasoning, and are critically important for the models to acquire world knowledge required in many problem-solving benchmarks [Zellers et al., 2019, Hendrycks et al., 2021, Wei et al., 2024], and even more specialized knowledge required for solving math and coding problems [Jain et al., 2025, Rein et al., 2023]. However, the dynamics of knowledge acquisition from training data are extremely complex, and various unintended biases [Zheng et al., 2024] and behaviors [Nasr et al., 2023] can arise, such as the "reverse curse" [Berglund et al., 2024] and increased hallucination [Gekhman et al., 2024]. Moreover, previous work also shows that in addition to learning generalizable knowledge, large language models (LLMs) can unintentionally memorize training text verbatim, which may then be extracted through simple prompting [Carlini et al., 2019, 2021, 2022]. This raises concerns about privacy, copyright, and the trustworthiness of AI systems. As a result, developing a better understanding of memorization in foundation models has become a central focus of recent research.

---

*Work done during an internship at Google.

39th Conference on Neural Information Processing Systems (NeurIPS 2025).

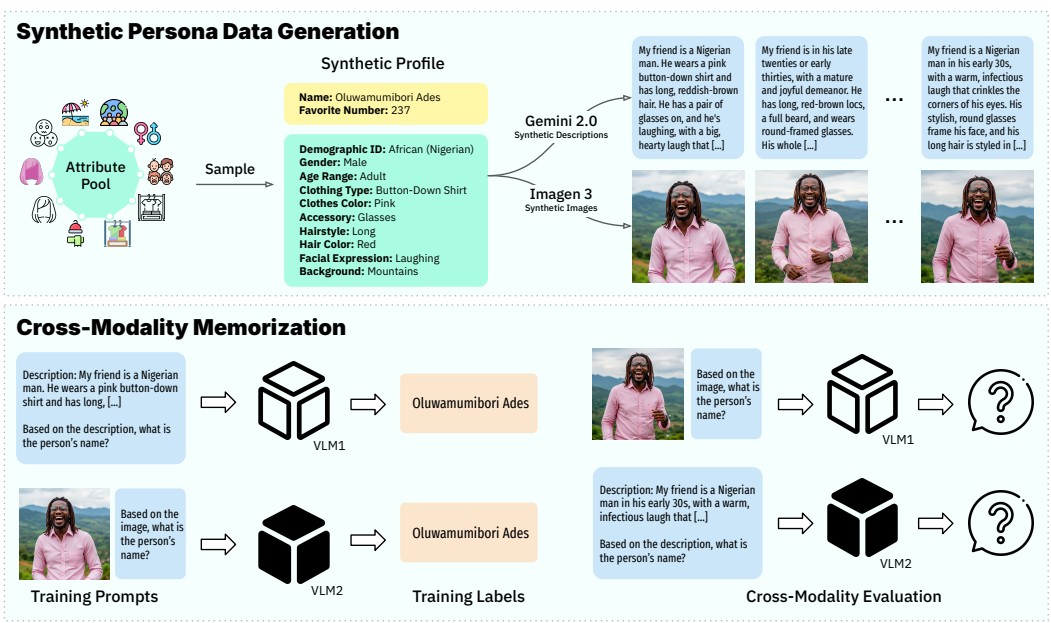

Figure 1: **Illustration of Our Data Generation Pipeline and Evaluation Pipeline.**

However, while prior work has investigated memorization within individual modalities—such as the regurgitation of specific training text sequences by LLMs [Carlini et al., 2019, 2021, 2022] and the reproduction of specific training images by diffusion models [Somepalli et al., 2023b, Carlini et al., 2023, Wen et al., 2024a]—these studies primarily focus on *unimodal* memorization. In contrast, solving complex real-world problems increasingly relies on multimodal models that are trained to process and integrate information from diverse media types, including text, images, audio, and video. This integration across modalities necessitates extending the study of memorization to account for *cross-modality* behaviors.

In this work, we address this gap by taking a first step toward systematically quantifying cross-modality memorization between text and images in vision-language models (VLMs). Under controlled training conditions, we examine how knowledge is memorized in one modality and transferred across modality boundaries to support inference in another.

Specifically, our investigation focuses on *factual knowledge memorization* across the visual and textual modalities in VLMs. Our task design is motivated by emerging applications of VLMs such as personal assistants, where models are often "personalized" using user-provided data such as emails, calendars, to-do lists, photos, and screenshots. For example, to assist with vacation planning, a model may check a calendar to identify suitable dates and examine past photos to suggest similar destinations. Gaining a clearer understanding of how factual knowledge is internalized and transferred across modalities is a critical step toward building robust and trustworthy multimodal personal agents.

To facilitate this study, we introduce the "persona" dataset, designed to systematically investigate cross-modal factual memorization, as shown in Figure 1. We begin by generating a diverse set of synthetic persona profiles, each featuring various attributes. For each persona profile, we create both textual descriptions and visual representations (e.g., synthetic photographs) to capture the person's appearance.

Next, we train vision-language models to recognize the person's name from a single modality, either the description or the image. During inference, we assess the model's ability to recall the person's name when presented with a held-out image or description. Our primary focus is on cross-modal knowledge transfer, examining the model's ability to answer textual questions when trained on visual data, and vice versa. To determine whether the model learns facts consistently across modalities, we compare the accuracy between the training and testing modalities.

Our systematic analysis reveals that facts learned in one modality are automatically transferred to the other. However, there is a noticeable performance gap between recalling information within the

original ("source") modality and recalling it in the "target" modality in many scenarios. Notably, this gap is asymmetric, with knowledge transfer from image to text being more effective than from text to image, even in larger and more capable models. A similar inconsistency arises in machine unlearning, where the model struggles to unlearn facts in a modality different from the one in which they were originally learned. Additionally, we identify a cross-modal "multi-hop curse" in multimodal models, where the model struggles to infer complex cross-modal relationships despite successfully learning simpler single-hop associations.

To better understand whether the cross-modality knowledge barriers are surface-level phenomena or more fundamental weaknesses in the standard learning setup, we further investigate whether simple mitigating strategies can improve cross-modal transferability. We find that using diverse training data and larger, more capable models can mitigate overfitting, but it does not improve the transferability rate. In contrast, augmenting training data with synthetic image-text pairs, generating images from text inputs and captions from image inputs, can effectively bridge the modality gap, particularly with in-distribution data augmentation. However, out-of-distribution data proves less beneficial in this case.

We hope our study contributes to a deeper understanding of cross-modal memorization phenomena, particularly within the context of VLMs. By highlighting the asymmetries in knowledge transfer between modalities, we aim to foster more robust, efficient, and privacy-conscious multimodal systems. Future work will focus on refining mitigation strategies and exploring additional modalities to generalize our findings further. We believe that addressing these challenges is crucial for developing AI systems that can safely and effectively integrate and process diverse types of data.

## 2 Related Work

Modern generative models memorize and regurgitate training data [Carlini et al., 2019, Inan et al., 2021, Zhang et al., 2023], presenting potential risks such as privacy leakage and copyright infringement [Henderson et al., 2023]. This phenomenon in language models has been extensively discussed in prior literature. Carlini et al. [2019] systematically examined such occurrences by injecting "canaries" into the training data. Later, Carlini et al. [2022], Kandpal et al. [2022] showed that model memorization correlates with model size, canary repetition frequency, and sequence length. Beyond verbatim copying, factual knowledge memorization has also been studied [Allen-Zhu and Li, 2025], wherein models recall specific facts present in their training datasets. This type of memorization is considered more aligned with realistic usage scenarios, as it often involves prompting the model with queries that differ from the exact training instances.

Research into memorization has also extended to image generation models [Somepalli et al., 2023b,a, Wen et al., 2024a]. Somepalli et al. [2023b] discovered that verbatim memorization also occurs in diffusion models, where the model can generate exact replicas of training images when prompted with corresponding training data inputs. Later, Carlini et al. [2023] demonstrated that, similar to language models, the extent of memorization in diffusion models is heavily dependent on model and data sizes. Analogous to factual knowledge memorization in language models, image generation models also memorize styles [Somepalli et al., 2024] and copyrighted characters [He et al., 2025]. In Jayaraman et al. [2024], Kokhlikyan et al. [2024], the authors investigate unintentional memorization in CLIP-style VLMs, whereas our work focuses on LLaVA-style VLMs.

## 3 Cross-Modality Factual Knowledge Memorization

### 3.1 Synthetic Persona Data

Injecting canaries is a widely adopted strategy in previous work [Carlini et al., 2019, Wen et al., 2024b] to prevent data contamination and maintain experimental control. Accordingly, we first develop a synthetic persona dataset as "canaries." Our synthetic persona dataset consists of a set of image-description samples representing synthetic profiles. Given that personal assistants represent a promising application for vision-language models, the setting for our synthetic persona dataset is chosen to simulate a scenario where the model learns to identify individuals (e.g., a user's acquaintances) based on images (similar to those saved on a user's phone) or or their textual descriptions (as might appear in text messages or emails).

An overview of our data generation pipeline is presented in Figure 1. In detail, the creation involves the following steps:

**I. Attribute Pool Definition:** The foundation of our dataset generation lies in defining a comprehensive set of attributes that constitute a persona. These attributes are categorized to cover various aspects of an individual's appearance, demographics, and contextual elements. For each category, we curate a pool of possible values, drawing inspiration from typical characteristics used in image generation and description. These attribute categories include:

- **Demographics:** 8 demographic identities, 2 genders, and 3 age ranges.
- **Visual Characteristics:** 7 clothing styles/types, 7 clothes colors, 7 accessories, 7 hairstyles, and 7 hair colors.
- **Contextual Elements:** 7 facial expressions and 12 background scenes.

The specific values within these pools are chosen to maximize diversity and verisimilitude, allowing for a large combinatorial space of unique profiles of

$$8 \times 2 \times 3 \times 7 \times 7 \times 7 \times 7 \times 7 \times 7 \times 12 = 67,765,824$$

combinations from the attribute pools. The complete list is provided in Appendix B.

**II. Profile Synthesis:** The synthetic profile synthesis process involves the following steps:

1. **Attribute Combination:** A unique combination of attributes is sampled without replacement from the defined pools to establish the core characteristics of a persona.
2. **Name Assignment:** A unique fictional name is generated, conditioned on the demographic identity and gender to maintain cultural relevance and consistency.
3. **PII Association:** To facilitate privacy-related experiments, we associate each synthetic character with PII data. Given that most models are well-aligned and consistently reject real PII data requests (e.g., Social Security Numbers), we opt to use "favorite number" as a surrogate. This favorite number is a randomly generated three-digit number.

**III. Image-Description Generation:** The generation of image-description pairs is a crucial step in creating a comprehensive dataset that effectively represents synthetic personas. This process not only ensures visual diversity but also enhances the contextual realism of each profile.

- **Image Generation:** Using the synthesized profiles (attribute combinations), we leverage a state-of-the-art text-to-image generation model Imagen 3 [Baldridge et al., 2024] to create realistic profile pictures.
- **Textual Description Generation:** To complement the generated images, we create rich textual descriptions for each persona profile picture. These descriptions go beyond merely listing attributes, instead presenting a natural, narrative, and contextually rich portrayal of the synthetic individual. We employ Gemini 2.0 [Team et al., 2023] to generate these textual descriptions, conditioned on both the generated image and the profile attributes. To enhance realism, we instruct the model to adopt a more conversational tone, similar to how a person would describe a friend.

Additionally, we generate multiple image-description pairs for each persona to simulate scenarios where a single individual has several images. This approach parallels the language model training process, where models are exposed to multiple paraphrases of a fact [Allen-Zhu and Li, 2025]. The generation prompts are provided in Appendix C.

**Final Dataset Composition:** The resulting synthetic persona dataset consists of a collection of 100 unique personas. Each persona is characterized by the following elements:

- A unique name.
- An associated favorite number.
- An associated set of specific attributes.
- A set of 100 image variants for training and 1 distinct image for testing.
- A set of 100 textual description variants for training and 1 distinct textual description for testing.

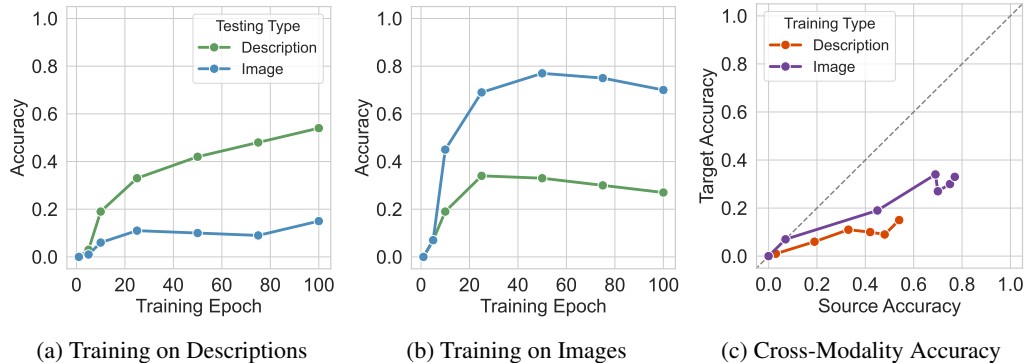

| (a) Training on Descriptions | (b) Training on Images | (c) Cross-Modality Accuracy |
|:---:|:---:|:---:|

Figure 2: **Training with Single Description/Image.** Factual memorization transfers between modalities, but there is a significant gap between the source and target modalities.

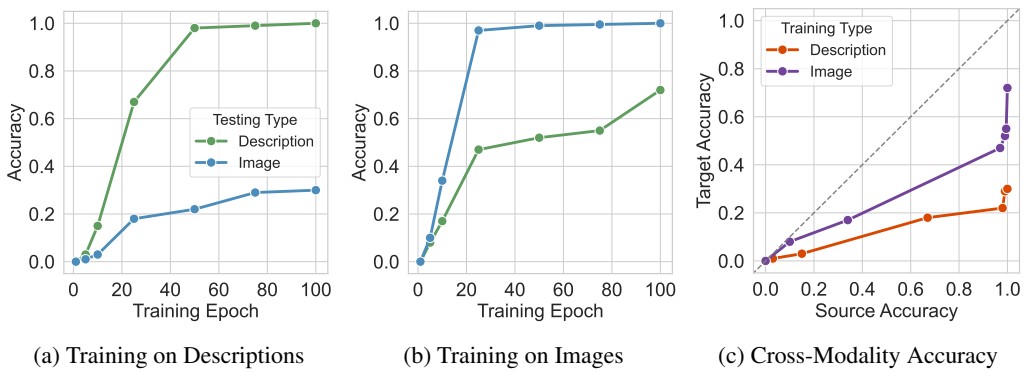

| (a) Training on Descriptions | (b) Training on Images | (c) Cross-Modality Accuracy |
|:---:|:---:|:---:|

Figure 3: **Training with Image/Description Variants.** Using diverse input variations during training helps mitigate overfitting, but does not improve the cross-modal transferability rate.

## 3.2 Experimental Setup

We fine-tune the latest open-source vision-language model, Gemma-3-4b [Team et al., 2025], to conduct all our experiments. During fine-tuning, we utilize LoRA [Hu et al., 2022] with a rank of $r = 32$, a scaling factor of $\alpha = 32$, and a dropout probability of $0.05$. We use AdamW [Loshchilov and Hutter, 2017] with a learning rate of $2 \times 10^{-4}$ and a batch size of $16$. All training is performed on a single Nvidia A100-80G GPU.

We employ two distinct training settings: one where the model is trained solely on textual descriptions and another where it is trained exclusively on images. In each setting, the input prompt consists of either a description or an image, followed by a question about the person's name. We train the model to accurately predict the associated name based on the given input.

During testing, we evaluate the model's recognition accuracy by asking the same question in both modalities separately, assessing the model's ability to recall the correct name in each modality. Note that we test on seen persona profiles, but the specific test inputs (image or textual descriptions) are held out and never seen verbatim by the model during training. In our experiments, we deliberately train for an extreme number of epochs to simulate a worst-case privacy scenario, where memorized signals are not diluted by non-canary data. This setup provides a stress test for evaluating the robustness of potential mitigation methods.

## 3.3 Memorization Results

**Training with Single Description/Image:** We start with fine-tuning the model on a single image or description for each persona across multiple epochs. Specifically, we choose 1, 5, 10, 25, 50, 75, and 100 epochs. We report the recognition accuracy for both models trained on descriptions and

images in Figure 2a and Figure 2b, respectively, showing the test accuracy on held-out descriptions and images.

In the description training setting, the model's accuracy for both modalities consistently improves as training progresses. In contrast, training on images tends to lead to overfitting after 25 epochs, as the model's performance decreases afterwards.

Although cross-modal knowledge transfer occurs in both training scenarios, there remains a significant performance gap between the training modality and the testing modality. This gap is more evident in Figure 2c, where the plot compares the accuracy between the training (source) modality and the testing (target) modality. Ideally, a perfect model would align with the dotted line, indicating no discrepancy between the two modalities. However, both training settings reveal a noticeable gap. This disparity underscores the difficulty of transferring learned representations between text and image domains.

> **Takeaway 1:** Factual memorization transfers between modalities, but there is a significant gap between the source and target modalities.

**Training with Image/Description Variants:** To mitigate the effect of overfitting associated with training on a single description or image for multiple epochs, we employ a different strategy in this experiment. Instead of repeatedly using the same input, we use a new description or image for each epoch. This approach is analogous to factual knowledge learning with paraphrased texts, which has been shown to help models generalize more effectively [Allen-Zhu and Li, 2025]. By introducing variations during training, the model avoids memorizing a fixed representation and instead learns the actual knowledge.

As depicted in Figure 3, training with diverse inputs significantly improves the test accuracy compared to the single-input scenario. In Figure 3a, where the model is trained on descriptions, the accuracy reaches nearly perfect levels, suggesting that the model benefits from varied textual data. Similarly, in Figure 3b, when trained on images, the model also achieves high accuracy without the early saturation and overfitting observed previously. These results highlight that varying the input per epoch effectively prevents overfitting and allows the model to generalize better across training instances.

However, despite these improvements in accuracy, the cross-modal transferability remains limited, as shown in Figure 3c, where the slope remains similar to the single-sample training scenario. This indicates that, while input variability mitigates overfitting, it does not necessarily enhance the model's ability to bridge the gap between different modalities.

> **Takeaway 2:** Using diverse input variations during training helps mitigate overfitting, but does not improve the cross-modal transferability rate.

One interesting observation is that the target accuracy across the modality continues to improve (especially for the image ⇒ text scenario) after the source modality saturates at near-perfect accuracies. This emphasizes the importance of diverse training samples on learning robust knowledge representations that go beyond what single-modality benchmarks can typically show.

**Training with Model Sizes:** As previous work [Carlini et al., 2022] indicates, larger models tend to memorize information more easily. To investigate the impact of model size on cross-modal transferability, we train three models from the Gemma3 family: Gemma3-4B-IT, Gemma3-12B-IT, and Gemma3-27B-IT. As shown in Figure 4, increasing the model size consistently improves the accuracy for both source and target modalities. However, despite the improvements in both individual modality accuracy and cross-modality accuracy as the model size increases, the rate of cross-modal transferability, indicated by the slope of the lines in the accuracy plots, remains nearly unchanged. This suggests that while larger models are more effective in learning within each modality, they do not necessarily enhance the transfer of learned information between modalities.

These findings imply that model size primarily contributes to improved memorization and recognition within a single modality rather than facilitating cross-modal generalization. Consequently, even as the model capacity increases, bridging the gap between image-to-text and text-to-image transfer remains a challenging task.

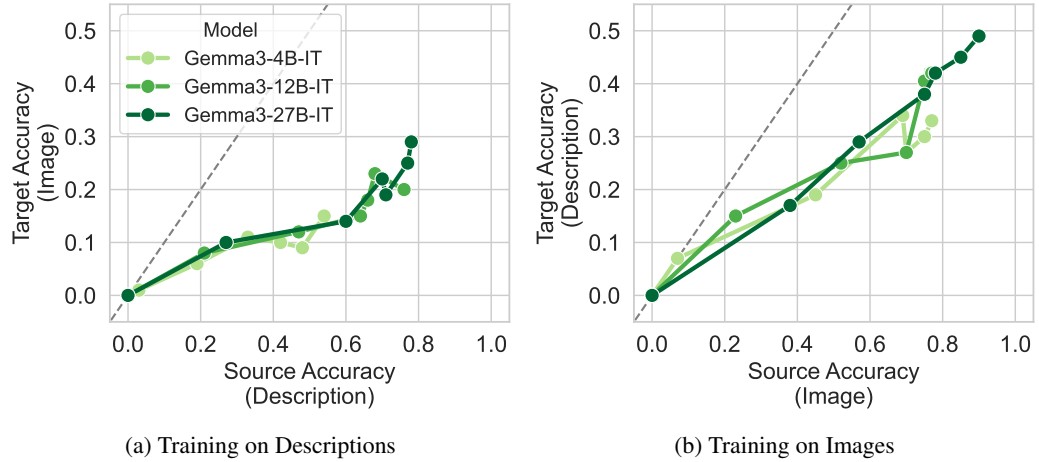

(a) Training on Descriptions        (b) Training on Images

Figure 4: **Training with Different Model Sizes.** Increasing model size enhances accuracy within each modality, but maintains a similar cross-modal transferability rate.

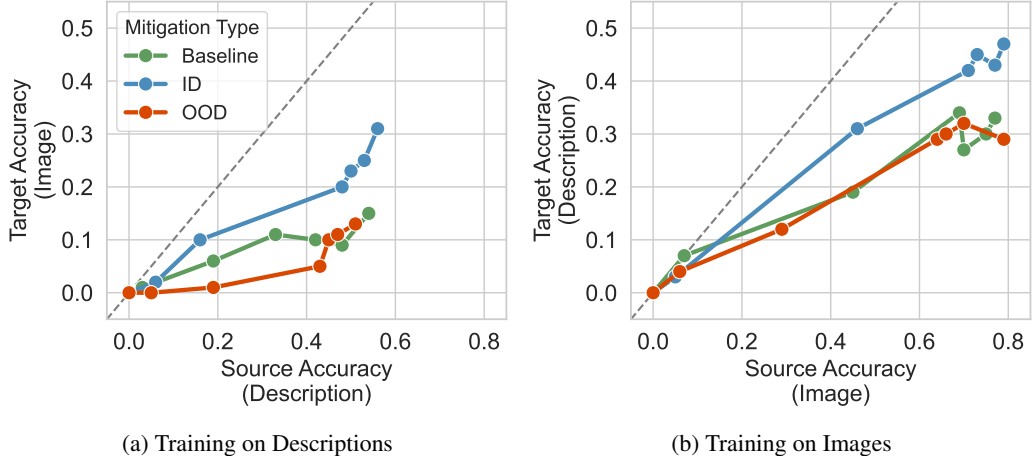

(a) Training on Descriptions        (b) Training on Images

Figure 5: **Training with Augmentation Mitigation.** Augmenting training data with in-distribution image-caption samples is essential for enhancing cross-modal transferability, while out-of-distribution data does not provide similar benefits.

> **Takeaway 3:** Increasing model size enhances accuracy within each modality, but maintains a similar cross-modal transferability rate.

### 3.4 Mitigation with Image-Caption Augmentation

To further enhance cross-modal transferability, we propose a simple yet effective approach that augments the training data with synthetic image-text pairs. Specifically, for a given description or text input, we generate a corresponding image conditioned on the text. Conversely, for an image input, we generate a descriptive caption. These generated pairs are introduced as captioning data, where the model is asked to describe an image.

In this experiment, we utilize two types of data for mitigation: 1) in-distribution (ID) data, where we augment the training set with held-out synthetic persona images and descriptions that closely resemble the training data, and 2) out-of-distribution (OOD) data, where we introduce incorporate images and captions from the COCO dataset [Lin et al., 2014]. This combination allows us to investigate the impact of data diversity on cross-modal transferability.

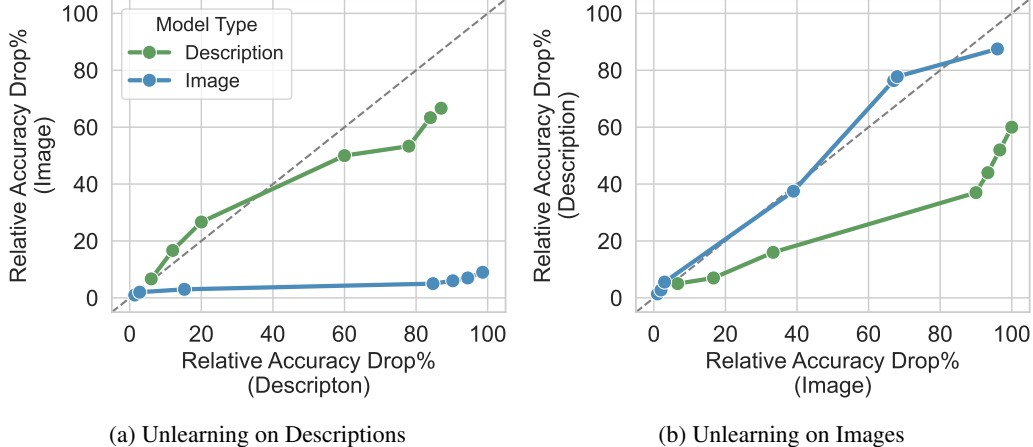

(a) Unlearning on Descriptions        (b) Unlearning on Images

Figure 6: **Unlearning cross Modalities.** Unlearning effects are modality-dependent: when unlearning happens in the same modality as learning, the effect remains relatively consistent across both modalities. In contrast, when unlearning occurs in a different modality from where the fact was originally learned, the effect does not transfer equally between modalities.

As shown in Figure 5, incorporating in-distribution (ID) data significantly improves cross-modal transferability, as indicated by the blue line consistently appearing above the baseline green line in both description and image training scenarios. In contrast, augmenting with out-of-distribution (OOD) data does not provide similar benefits, with the orange line remaining close to the baseline. This suggests the importance of augmenting training data within the same distribution to effectively improve transferability.

> **Takeaway 4:** Augmenting training data with in-distribution image-caption samples is essential for enhancing cross-modal transferability, while out-of-distribution data does not provide similar benefits.

### 3.5 Cross-Modal Unlearning

Machine unlearning [Bourtoule et al., 2021] has gained popularity as a method for removing specific user data without requiring the model to be retrained from scratch. Consequently, it is essential to investigate how unlearning operates across different modalities.

A commonly used unlearning method involves applying gradient ascent on the forget dataset [Maini et al., 2024]. However, our empirical experiments reveal that this approach is challenging to fine-tune and significantly compromises model performance. Therefore, to make the controlled experiments more comparable with fresh knowledge learning results, we adopt a practically simpler workaround: to unlearn the association of a person (image or textual description) to a name, we generate a modified training where each profile is associated with completely different names, and (continue) training the model on this modified dataset. We then measure the model's accuracy (drop) of association with the original name as the unlearning performance. This allows us to reuse the same training setups, making the cross-modality measurements more easily comparable with previous results.

In detail, we start with the fine-tuned description model and image model from Section 3.3. Since we have two models (the fine-tuned description model and the fine-tuned image model) and two unlearning datasets (description-based and image-based), we systematically evaluate all possible combinations, which results in four experimental setups.

In Figure 6a, we present unlearning experiments with relative accuracy drops in both modalities when unlearning on description data. For the description model, the unlearning effect on both modalities exhibits a similar pattern (as indicated by the green line closely following the $y = x$ line). Interestingly, for image models, unlearning on description data predominantly affects the description test accuracy, while the image test accuracy drop remains quite small. In Figure 6b, we

Table 1: **Cross-Modal Multi-Hop Learning** Cross-modal models suffer from the multi-hop curse, where accurate performance on individual tasks does not translate to multi-hop reasoning.

| Training Data | | Test Type | | | | |
|---|---|---|---|---|---|---|
| **Base Data** | **Multi-Hop Data** | **Desc⇒Name** | **Image⇒Name** | **Desc⇒FN** | **Image⇒FN** | **Name⇒FN** |
| | Desc⇒FN | 1.00 | 0.35 | 1.00 | 0.31 | 0.08 |
| Desc⇒Name | Image⇒FN | 1.00 | 0.36 | 0.39 | 1.00 | 0.05 |
| | Name⇒FN | 1.00 | 0.36 | 0.11 | 0.05 | 0.99 |
| | Desc⇒FN | 0.68 | 1.00 | 1.00 | 0.47 | 0.07 |
| Image⇒Name | Image⇒FN | 0.62 | 1.00 | 0.50 | 1.00 | 0.07 |
| | Name⇒FN | 0.64 | 1.00 | 0.27 | 0.65 | 0.99 |

show unlearning experiments on image data. Similar to the description data pattern, for the image (source modality) model, the relative accuracy drop on both modalities is comparable. However, for the text model, the unlearning effect on description data is notably weaker, while the drop in image accuracy is more significant.

Overall, these observations reveal a gap in cross-modalities unlearning: when unlearning facts in a modality different from the one in which the model originally learned the fact, the unlearning effect does not transfer equally between modalities. In contrast, when unlearning occurs in the same modality where the fact was originally learned, the unlearning effect is relatively similar across both modalities. This underscores the importance of carefully designing unlearning training data and developing more effective unlearning techniques for cross-modal scenarios.

> **Takeaway 5:** Unlearning effects are modality-dependent: when unlearning happens in the same modality as learning, the effect remains relatively consistent across both modalities. In contrast, when unlearning occurs in a different modality from where the fact was originally learned, the effect does not transfer equally between modalities.

## 3.6 Cross-Modal Multi-Hop Learning

Large language models often exhibit the "multi-hop curse," where they fail to infer that *A is C* when the model is trained on *A is B* and *B is C* [Balesni et al., 2024]. In this section, we investigate this multi-hop scenario in a cross-modal context.

We define the original description or image mapping to name data points as *A is B* data (Desc⇒Name and Image⇒Name), while a three-digit number representing a person's favorite number (FN) serves as *C*. To investigate multi-hop reasoning, we introduce various data combinations involving base data: Desc⇒Name (given the description, predict the corresponding name) or Image⇒Name; multi-hop data: Desc⇒FN (given the description, predict the corresponding favorite number), Image⇒FN, and Name⇒FN. During testing, we assess whether the model can recall the person's favorite number given a description, image, or name as a cue.

As shown in Table 1, cross-modal models also exhibit the multi-hop curse. For instance, when the model is trained on Desc⇒Name and Name⇒FN data, it achieves perfect accuracy on both tasks individually. However, when directly prompted with a description to retrieve the favorite number, the accuracy drops drastically to only $11\%$, indicating a significant challenge in the multi-hop scenario. In contrast, the model trained on Image⇒Name exhibits a lower barrier, achieving over $50\%$ accuracy for Image⇒FN. Even when tested on Desc⇒FN, the accuracy is considerably higher compared to the model trained on descriptions, suggesting that image-based training better supports multi-hop reasoning.

> **Takeaway 6:** Cross-modal models suffer from the multi-hop curse, where accurate performance on individual tasks does not translate to multi-hop reasoning.

## 4 Conclusion and Limitations

Our study systematically investigates cross-modality memorization in vision-language models using a synthetic persona dataset. We uncover an asymmetric performance gap between source and target modalities, where knowledge transfer from image to text is more effective than from text to image. We also demonstrate that while augmenting training data with in-distribution synthetic image-text pairs helps bridge this gap, out-of-distribution data proves less effective. Furthermore, we identify challenges in cross-modal unlearning, as unlearning in one modality does not fully translate to the other. Additionally, we reveal that vision-language models suffer from the multi-hop curse, where the model struggles to infer complex cross-modal relations despite successful learning of simpler single-hop associations.

Our findings underscore the complexity of achieving robust cross-modal transferability and highlight the need for improved training strategies to enhance generalization across modalities. While this work focuses on text and image modalities, future research should explore additional modalities, develop more sophisticated unlearning techniques, and design approaches to better handle multi-hop reasoning. Addressing these challenges is crucial for building more reliable and privacy-conscious multimodal AI systems.

## Acknowledgments

Icons from https://www.flaticon.com/ were used to create the illustration figure in this paper.

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

# A   Broader Impacts

This paper systematically evaluates the state of cross-modal knowledge transfer in multimodal language models. Through extensive empirical studies, we reveal the existence of a significant knowledge transfer gap between vision and text domains. Despite this gap, our findings demonstrate that knowledge does transfer between these modalities, albeit asymmetrically.

Understanding this transfer is crucial for the future design and deployment of multimodal models, as it has important implications for both performance optimization and privacy preservation. Specifically, our results highlight the potential risk of unintended information leakage across modalities. For example, sensitive information learned from visual data could inadvertently influence text-based outputs and vice versa.

To mitigate such risks, researchers and practitioners should carefully evaluate cross-modal knowledge interactions, especially when designing models for real-world applications that handle sensitive or personal data. We encourage the community to further investigate robust techniques that ensure cross-modal privacy while maintaining model performance.

# B   Attribute Pool

- **Demographic Identity:** African (South African), Caucasian (American), Caucasian (British), East Asian (Chinese), East Asian (Korean), Hispanic (Mexican), Iberian (Spanish), Middle Eastern (Saudi)

- **Gender:** Female, Male

- **Age Range:** Adult, Middle Aged, Senior

- **Clothing Type:** Button-Down Shirt, Hoodie, Jacket, Polo, Sweater, T-Shirt, Tank Top

- **Clothes Color:** Black, Brown, Gray, Pink, Red, White, Yellow

- **Accessory:** Earrings, Glasses, Hat, Headphones, Jewelry, No Accessory, Scarf

- **Hairstyle:** Afro, Bald, Curly, Dreadlocks, Long, Medium, Short

- **Hair Color:** Black, Blonde, Brown, Gray, Pink, Red, White

- **Facial Expression:** Angry, Frowning, Laughing, Neutral, Sad, Smiling, Surprised

- **Background:** Beach, Cafe, Cityscape, Forest, Graffiti Wall, Library, Mountains, Office, Plain Black, Plain Grey, Plain White, Sports Field

# C   Data Generation Prompts

---

**Image Generation Prompt for Imagen 3**

Generate a highly detailed, photorealistic photo of a person. Key characteristics include:
Demographic Identity: African (Nigerian)
Gender: Male
Age Range: Adult
Clothing Type: Button-Down Shirt
Clothes Color: Pink
Accessory: Glasses
Hairstyle: Long
Hair Color: Red
Facial Expression: Laughing
Background: Mountains

---

> **Description Generation Prompt for Gemini 2.0**
>
> <image>
>
> Describe the person in the image in a detailed and natural way, making it easy for someone to recognize them based on the description. Write as if you are describing a friend of my. Ensure the description flows naturally and includes the following attributes:
> Demographic Identity: African (Nigerian)
> Gender: Male
> Age Range: Adult
> Clothing Type: Button-Down Shirt
> Clothes Color: Pink
> Accessory: Glasses
> Hairstyle: Long
> Hair Color: Red
> Facial Expression: Laughing
> Background: Mountains
>
> Feel free to add any additional distinguishing features that enhance the portrayal. Please provide the description directly, without extra formatting or instructions.

## D  Training and Testing Prompts

> **Training/Testing Prompt for Image Data**
>
> <image>
>
> Based on the image, what is the person's name?

> **Training/Testing Prompt for Description Data**
>
> Description: My friend is a Nigerian man. He wears a pink button-down shirt [. . . ]
>
> Based on the description, what is the person's name?

## E  Test Prompt Engineering

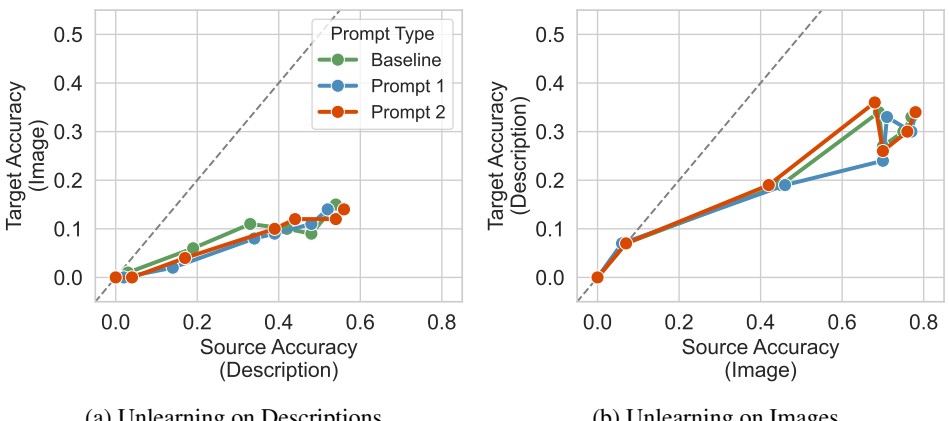

(a) Unlearning on Descriptions          (b) Unlearning on Images

Figure 7: **Testing with Different Prompt Strategies.**

At the beginning of the project, we tested a naive mitigation strategy by appending modality transfer-aware prompts during inference. We evaluated two prompts:

- Prompt 1: Try to recall knowledge learned from another domain.
- Prompt 2: Given this description/image, think about what you learned from the image/description.

As shown in Figure 7, neither prompt leads to a significant change in transferability.

## F    Unintentional Memorization in VLMs

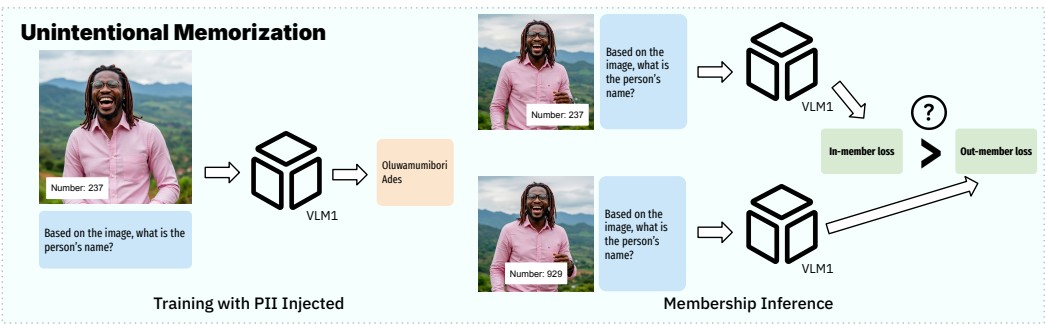

Figure 8: **Illustration of Unintential Memorization.**

At the end, we share a phenomenon observed in our experiments, which we refer to as unintentional memorization: the model inadvertently memorizes the injected PII from the image, despite it being entirely unrelated to the current task. As illustrated in Figure 8, we inject a "favorite number" into the image during training. Since the model is aligned to avoid generating contents related to potentially sensitive information such as social security numbers, in order to decouple the question of alignment (refusing to answer) and memorization, we use the "favorite number" as a surrogate for real PII in our measurements. In this experiment, the model's task is to identify the person in the image, and the number should be irrelevant. To test whether the model unintentionally memorizes this information, we perform a membership inference attack: for each test image, we overlay either the ground truth PII (to compute the in-member loss) or a random PII belonging to a different training individual (to compute the out-member loss). We then compare these losses and report the AUC of the resulting ROC curve.

We present the results in Figure 9. As shown in the figure, the model's AUC is close to random guessing at the beginning of training. However, as training progresses, the AUC steadily increases, reaching around 70% after 100 epochs. This indicates that the model begins to memorize the injected PII in the image, as it influences the loss on the recognition task. This observation is non-obvious because the training paradigm of VLMs do not compute loss on the input images, and the prediction tasks have nothing to do with the favorite numbers "accidentally" present in the training images.

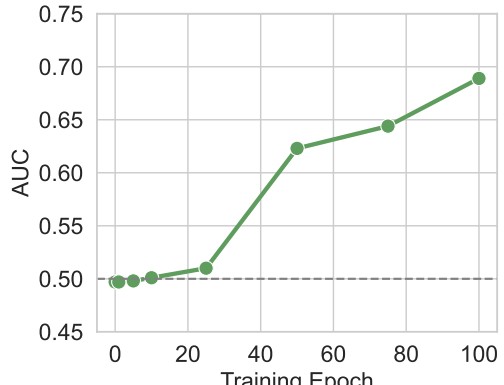

Figure 9: **Membership Inference Results for Unintentional Memorization**

We also attempted direct extraction of the PII, but the success rate was near zero in all cases. In contrast, membership inference reveals a much stronger signal of memorization.

We argue that this type of unintentional memorization poses significant privacy risks and warrants deeper investigation in future work. For instance, in the context of training LLM-based agents

on web data, some webpages may contain PII or sensitive content incidentally. Even if such content is unrelated to the task, the model may still inadvertently memorize it. As a result, future efforts should focus on carefully filtering or redacting training data to mitigate these risks.

> **Takeaway 7:** VLMs can unintentionally memorize irrelevant visual PII. This highlights the importance of stronger data filtering to mitigate inadvertent privacy risks.

