# OpenReview forum: "Quantifying Cross-Modality Memorization in Vision-Language Models"
_NeurIPS.cc/2025/Conference — NeurIPS 2025 poster_

### Official Review · Reviewer_r17s · 2025-06-19

**Clarity:** 3
**Significance:** 3
**Originality:** 2
**Rating:** 4
**Confidence:** 3

**Summary:**

The paper studies cross-modality memorization in vision-language models for sensitive attribute prediction task on a synthetic dataset introduced by the authors. The paper quantifies memorization in terms of predicting correct attribute in both source and target modalities. Here the source is the modality that is used for training the model and target is the modality that is used only for inference.
The paper suggests that memorization transfers between modalities but it is significantly lower in the target modality. Lastly, the paper proposes mitigations using data augmentation and studies cross-modality unlearning.

**Questions:**

+  The paper first mentions that there are ~64B profiles but then later that there are only 100 unique profiles. What are the differences between profile and persona ? How large is the final dataset ?
+ How did you ensure that the dataset is diverse and representative enough for the study ?
+ How much data was added during training with variants ? How did you know how many variants you need to add ?

**Ethical Concerns:**

["NO or VERY MINOR ethics concerns only"]

**Limitations:**

The paper discussed future work but does not discuss the limitations of the work.

**Paper Formatting Concerns:**

No paper formatting concerns

**Quality:**

2

**Strengths And Weaknesses:**

**Pros**

+ The paper is clearly and well written it conducts thorough experiments for different hyper parameters such as model size, increasing data diversity to avoid overfitting.
+ The paper proposes a mitigation for the cross modality transferability gap and shows that proposed mitigation helps to close the gap.
+ It’s also interesting to see multi-hop unlearning results for cross-modality dataset.

**Cons**
Cons
+ The experimental evidence is provided only for the benchmark proposed by the paper. It is unclear whether the observations made by the authors are generalizable cross other datasets and modalities.
It could be that the observations made in the paper reflect some of the properties in the dataset. It might be good to try the experiments on other attribute prediction dataset and task.
+  It seems that the authors use the COCO dataset as OOD dataset in one of their experiments. It might have been good to run all experiments on COCO dataset instead of the synthetic dataset and use synthetic dataset as OOD. It would be interesting to see whether the claims made by the authors will hold for this setup too.
+ In terms of VLM memorization there is this other paper that is not cited
https://arxiv.org/abs/2402.02103

---

> ### Author Rebuttal · Authors · 2025-07-31
>
> We sincerely appreciate your valuable feedback and the time you've dedicated to providing it. Below, we address specific points you raised:
>
> > It is unclear whether the observations made by the authors are generalizable cross other datasets and modalities.
>
> This point is also mentioned in our limitations (Section 4). Our broad goal is to demonstrate that the cross-modal memorization findings we discovered can emerge, and they are possible to happen in other modalities. Therefore, we hope our work can serve as a starting point for fellow researchers to study memorization in different modalities.
>
> > It might have been good to run all experiments on COCO dataset instead of the synthetic dataset and use synthetic dataset as OOD.
>
> Using COCO will be more realistic, but we use fully synthetic data to prevent data contamination. With real images, it is often unclear whether the associated knowledge has already appeared during pretraining, making it difficult to isolate cross-modal memorization effects. Our synthetic setup enables a controlled investigation of memorization without the confounding risk of training data leakage. Additionally, using real human faces might raise serious privacy and consent concerns.
>
> > In terms of VLM memorization there is this other paper that is not cited.
>
> Thank you for pointing out this paper on CLIP models. We will include a discussion of it in the camera-ready version.
>
> > The paper first mentions that there are ~64B profiles but then later that there are only 100 unique profiles. How did you ensure that the dataset is diverse and representative enough for the study?
>
> The total number of possible attribute combinations is approximately 64 billion, which reflects the size of the design space. We included this to emphasize the diversity and richness of the attribute pool. From this large space, we sample 100 unique profiles to construct a diverse and representative dataset while keeping the experimental setup manageable.
>
> > How much data was added during training with variants? How did you know how many variants you need to add?
>
> The number of variants depends on the target number of training epochs. For example, if we aim for 20 epochs, we add 19 additional variants per profile, alongside the original, resulting in 20 distinct samples. Then, we perform a single training epoch, where each variant is seen once—effectively simulating a 20-epoch training schedule with different variants.
>
> We hope our response can resolve your concerns regarding our paper. Please let us know if you have any more questions.

---

> > ### Comment · Reviewer_r17s · 2025-08-04
> > **Response to the authors**
> >
> > Thank you authors for responding to my questions. I went over the answers and the paper again. I'll leave my score unchanged.

---

### Official Review · Reviewer_Rdda · 2025-06-26

**Clarity:** 3
**Significance:** 3
**Originality:** 3
**Rating:** 4
**Confidence:** 4

**Summary:**

This paper carries out studies of cross-modal knowledge extraction in a multimodal LLM. To do this, they introduce a new dataset of personas, consisting of synthetically generated images of (fake) people, together with their textual descriptions. The MLLMs are then fine-tuned with this information in one modality (image or text), and extraction is attempted from the other. The main finding of the paper is that image to text extraction is significantly easier than text to image extraction.

**Questions:**

n/a

**Ethical Concerns:**

["NO or VERY MINOR ethics concerns only"]

**Final Justification:**

After discussion I think this is a solid paper and I am supportive of acceptance.

**Quality:**

4

**Strengths And Weaknesses:**

Strengths:

+ Overall I thought this is a very solid paper -- the dataset is well-made and interesting, the experiments are quite well-done, the conclusions are presented crisply. Execution-wise this is a really strong paper.

Weaknesses:

- To me, the main weakness of this paper is in the way is it framed. The current framing of the paper suggests that the results have serious implications for privacy, showing that models memorize private information, and that this memorization could be "cross-modal".

However usually memorization in the the privacy/memorization literature refers to incidental memorization where there is a very small amount of data about a single person in a very large crowd, such as a single data point in a large crowd. My understanding is that the setting of the current paper is a little bit different in a subtle way -- since we are directly fine-tuning with each person's data for many epochs; in my view, this paper's setting is closer to the Physics of LLMs: Knowledge Extraction (https://arxiv.org/abs/2309.14316) paper but for multimodal models. This is not to say that the results are uninteresting -- in fact they are extremely well-done and interesting -- but the implications for privacy are perhaps not quite there.

- Minor: It's also not quite correct that there has no work at all on multimodal memorization -- [1, 2] look at multimodal memorization in a different type of multimodal (CLIP) models.

[1] https://arxiv.org/abs/2402.02103
[2] https://arxiv.org/abs/2504.05651

---

> ### Author Rebuttal · Authors · 2025-07-31
>
> We sincerely appreciate your valuable feedback and the time you've dedicated to providing it. Below, we address specific points you raised:
>
> > To me, the main weakness of this paper is in the way is it framed.
>
> We understand the concern. While prior work often focuses on unintentional memorization from large-scale pretraining, such experiments are extremely expensive to conduct—especially for VLMs, where training costs are significantly higher than for LLMs due to the length of visual tokens.
>
> Our experimental setting represents a worst-case privacy scenario, where the memorized signals are not diluted by non-canary data. This setup is particularly useful for stress-testing mitigation methods. We will revise our wording in the camera-ready version to improve our clarity.
>
> > It's also not quite correct that there has no work at all on multimodal memorization.
>
> Thank you for pointing out these two papers on CLIP models. They are indeed closely related to our work, and we will include a discussion of them in the camera-ready version.
>
> We hope our response can resolve your concerns regarding our paper. Please let us know if you have any more questions.

---

> > ### Comment · Reviewer_Rdda · 2025-08-04
> > **Thanks**
> >
> > Thank you for the response; if you could include a discussion of these points in the paper, then I will be supportive of acceptance.

---

> > > ### Author Response · Authors · 2025-08-04
> > > **Response by Authors**
> > >
> > > We appreciate your feedback. We will incorporate a more detailed discussion of the points raised during the rebuttal in the camera-ready version, where we will have an additional page to do so.

---

### Official Review · Reviewer_F4yj · 2025-06-30

**Clarity:** 4
**Significance:** 3
**Originality:** 3
**Rating:** 5
**Confidence:** 3

**Summary:**

This paper explores the memorization behavior in a multi-modal (image-text) foundation model using a novel synthetic dataset that is easy to control. Carefully designed experiments test the model’s ability to recognize facts in both modalities, which were only presented in one modality during training. The findings reveal that knowledge does not fully transfer from the source modality to the target modality, and larger model sizes, nor diverse training variations can bridge the gap. Training with more in-distribution image-caption pairs, on the other hand, does help. Finally, the paper reports asymmetries in the results depending on which modality was used as the source modality (e.g., for cross-modality transfer, effects of machine unlearning and the multi-hop curse).

**Questions:**

My first impression was that it was already a strong paper. To consider increasing an already good score, I would want to see how the conclusions generalize to other datasets/models/modalities and get more insights on causes or mitigation strategies.

Some minor suggestions:
- it could be nice to have the take-home in the figure caption. I found myself referring back to the text to check my interpretation of the figure.
- this might be personal taste, but I was comparing graph-lines between figures and it would have helped me if the xlims and ylims were consistent across figures. And to have it pointed out which lines were identical to one from a previous figure (same condition).

**Ethical Concerns:**

["NO or VERY MINOR ethics concerns only"]

**Final Justification:**

My questions/comments have been adequately addressed and after also considering the other reviews, I am still of the opinion that this is a solid paper. I'd recommend acceptance.

**Limitations:**

yes

**Paper Formatting Concerns:**

No concerns

**Quality:**

3

**Strengths And Weaknesses:**

I will list perceived strengths and weaknesses regarding Quality [Q], Clarity [C], Significance [S}, and Originality [O] below.

# Strengths
1. [C] The paper is beautifully written and very well structured. The introduction makes the aims and contributions clear from the start, and lays out nicely what the reader can expect (and what not to). The take-home boxes are very helpful too. The type of language used (not overly technical) makes it accessible to non-experts.
2. [Q] The experiments are carefully designed to address specific questions and they delivered clear take-home messages. This was made possible through a novel synthetic dataset that offers good experimental control.
3. [O] The custom synthetic dataset makes for an original addition to the work and describes a way for others to build their own.
4. [S] I can see this work being an important first steps for many follow-up papers to build on, speaking to many members in the NeurIPS community. Especially given the far-reaching impact of multi-modal foundation models, understanding their behavior and mitigating privacy and other memorization-related concerns is a valuable endeavor.

# Weaknesses
1. [Q] The paper is very engaging, which made me want to know more about potential causes for the observed patterns, something that wasn't touched upon very much.
2. [Q] In a similar vein, I would have loved to see more results on other mitigation strategies, although it was clearly acknowledged in the introduction that this was for future work. Could it not have been one big paper?
3. [S] The results are based on one (specialized) dataset and one model family (the focus on solely image-text modalities was already acknowledged in the paper).

---

> ### Author Rebuttal · Authors · 2025-07-31
>
> We sincerely appreciate your valuable feedback and the time you've dedicated to providing it. Below, we address specific points you raised:
>
> > The paper is very engaging, which made me want to know more about potential causes for the observed patterns, something that wasn't touched upon very much.
>
> Thank you for the positive feedback and your interest in the underlying mechanisms. We agree that understanding the causes of cross-modal asymmetries is a valuable and exciting direction for future work. Thoroughly investigating these causes may require a different set of experiments. We would like to focus on this in our follow-up work.
>
> > In a similar vein, I would have loved to see more results on other mitigation strategies, although it was clearly acknowledged in the introduction that this was for future work. Could it not have been one big paper?
>
> We tested out straightforward baseline methods in this paper. However, designing mitigation methods that can truly close the modality gap likely depends on first understanding the underlying causes—as you noted above. This would require a systematic and non-trivial analysis. Rather than overloading a single paper, we felt it was more appropriate to separate these efforts and ensure each component received the depth it deserves. We see this as an important next step and appreciate your encouragement in that direction.
>
> > The results are based on one (specialized) dataset and one model family (the focus on solely image-text modalities was already acknowledged in the paper).
>
> We have also run additional experiments with LLaVA 1.5. Due to rebuttal format constraints, we are unable to include full results and plots here, but we will include them in the camera-ready version. For example, for training with image/description variants scenario with 50 epochs, we observe the following source → target accuracy:
>
>
> |                          |    Gemma   |    LLaVA   |
> |:------------------------:|:----------:|:----------:|
> | Training on Descriptions | 0.98→0.22 | 0.97→0.15 |
> |    Training on Images    | 0.99→0.52 | 0.98→0.37 |
>
> We observe a similar asymmetric cross-modal transfer pattern. However, LLaVA exhibits slightly lower transferability in both directions compared to Gemma-3. In the future version, we plan to include more model families.
>
> More broadly, one of our goals is to demonstrate that a cross-modal memorization gap emerges even with state-of-the-art open-source VLMs like Gemma-3. We hope our pipeline serves as a useful evaluation routine for assessing newly developed multimodal models.
>
> > Minor suggestions
>
> Thank you for your helpful suggestions to improve the clarity of our paper. We will incorporate them in the camera-ready version.
>
> We hope our response can resolve your concerns regarding our paper. Please let us know if you have any more questions.

---

> > ### Comment · Reviewer_F4yj · 2025-08-03
> >
> > Thank you for taking the time to respond to the points/questions I raised and for providing additional empirical results. I went over the other reviews and rebuttals and I'm still of the opinion that this is a solid paper. My recommendation is still to accept it.

---

### Official Review · Reviewer_4B94 · 2025-07-03

**Clarity:** 3
**Significance:** 2
**Originality:** 3
**Rating:** 4
**Confidence:** 3

**Summary:**

This work takes the first step toward systematically quantifying cross-modal memorization in vision–language models. Rather than evaluating on arbitrary real-world data, it introduces a synthetic “persona” benchmark of paired images and text captions labeled with unique names. By fine-tuning VLMs to predict names from one modality and testing on the other, it uncovers a persistent, asymmetric performance gap and a multi-hop inference failure in cross-modal settings.

**Questions:**

My main concerns are in the method section, please see the weakness above.
Since this work is interesting to me, I would be willing to raise my score if the mentioned weaknesses are addressed in the rebuttal.

**Ethical Concerns:**

["NO or VERY MINOR ethics concerns only"]

**Final Justification:**

Thanks for authors' detailed and timely rebuttal. All of my questions have been answered with good answers and extensive qualitative results, and I am happy to raise my score.

**Limitations:**

The limitations of this study is not discussed in the paper.

**Quality:**

3

**Strengths And Weaknesses:**

Strengths:
1. The paper tackles an important and understudied problem by systematically quantifying how factual knowledge learned in one modality transfers to another in vision-language models.
2. The synthetic persona dataset is carefully designed with controlled image–text pairs, enabling clear attribution of memorization and transfer effects.
3. Experiments are thorough which covering single-input vs. variant training, model scaling, data augmentation, unlearning, and multi-hop reasoning, providing a comprehensive empirical picture.
4. Clear asymmetries (image-to-text and text-to-image transfer) and failure modes (overfitting, multi-hop curse, unlearning gaps) are identified, offering concrete takeaways for future methods.

Weaknesses:
1. Reliance on fully synthetic personas may limit external validity, as real-world data distributions are not evaluated.
2. The study focuses solely on name-recall tasks, leaving open whether richer or higher-level knowledge (e.g., events, attributes beyond names) exhibits similar cross-modal behavior.
3. All experiments use a single VLM family (Gemma-3-IT), so conclusions may not generalize to other architectures or pretraining regimes.
4. The impact of synthetic data quality (artifacts from Imagen 3 or Gemini 2.0) on memorization and transfer is not examined, risking biased results.
5. The limitations of this study is not discussed in the paper.

---

> ### Author Rebuttal · Authors · 2025-07-31
>
> We sincerely appreciate your valuable feedback and the time you've dedicated to providing it. Below, we address specific points you raised:
>
> > Reliance on fully synthetic personas may limit external validity, as real-world data distributions are not evaluated.
>
> We agree that using real images would be ideal. However, we use fully synthetic data to prevent data contamination and ensure annotation accuracy. With real images, it is often unclear whether the associated knowledge has already appeared during pretraining or in other data points, making it difficult to isolate cross-modal memorization effects. Our synthetic setup enables a controlled investigation of memorization without the confounding risk of training data leakage. Additionally, using real human faces might raise serious privacy and consent concerns.
>
> To minimize bias and ensure high data quality, we intentionally selected the state-of-the-art generative models in our paper. We used Imagen 3 for image generation for its top-tier performance in photorealism and content diversity (https://lmarena.ai/leaderboard/text-to-image/pre-generated-prompts-surprise-me-only). In contrast, alternatives like Midjourney or GPT-4o’s native generation often exhibit stylistic biases. For textual descriptions, we used Gemini 2.0, conditioning on structured persona attributes to significantly reduce hallucinations and maintain consistency across modalities.
>
> > The study focuses solely on name-recall tasks.
>
> We focused on the name-recall task because it serves as a straightforward proxy for factual memorization that is easy to verify, allowing us to minimize confounding variables during data generation and training. Additionally, we included results in the paper on recalling a “favorite number” as a secondary fact to demonstrate generalization beyond names.
>
> We agree that extending the study to include more abstract or higher-level knowledge—such as events or relationships—would provide valuable insights. Designing such experiments requires a different pipeline for data construction, and we view this as a promising direction for future work.
>
> > All experiments use a single VLM family (Gemma-3-IT), so conclusions may not generalize to other architectures or pretraining regimes.
>
> We have also run additional experiments with LLaVA 1.5. Due to rebuttal format constraints, we are unable to include full results and plots here, but we will include them in the camera-ready version. For example, for training with image/description variants scenario with 50 epochs, we observe the following source → target accuracy:
>
> |                          |    Gemma   |    LLaVA   |
> |:------------------------:|:----------:|:----------:|
> | Training on Descriptions | 0.98→0.22 | 0.97→0.15 |
> |    Training on Images    | 0.99→0.52 | 0.98→0.37 |
>
> We observe a similar asymmetric cross-modal transfer pattern. However, LLaVA exhibits slightly lower transferability in both directions compared to Gemma-3. In the future version, we plan to include more model families.
>
> More broadly, one of our goals is to demonstrate that a cross-modal memorization gap emerges even with state-of-the-art open-source VLMs like Gemma-3. We hope our pipeline serves as a useful evaluation routine for assessing newly developed multimodal models.
>
> > The limitations of this study is not discussed in the paper.
>
> We appreciate the reviewer pointing this out. While the limitations are discussed at the end of Section 4, we agree that they could be more clearly highlighted. Specifically, we acknowledged two main limitations:
> (1) Pur study focuses solely on image-text cross-modal knowledge transfer, whereas other modalities (e.g., audio, video) remain unexplored; and
> (2) We only evaluated baseline mitigation strategies, leaving more advanced architectural or algorithmic approaches to future work.
>
> To improve clarity, we will add a dedicated "Limitations" subsection title in the camera-ready version to make these points more explicit and accessible.
>
> We hope our response can resolve your concerns regarding our paper. Please let us know if you have any more questions.

---

> ### Comment · Reviewer_4B94 · 2025-08-08
> **Rebuttal Acknowledged**
>
> Thanks for authors' detailed and timely rebuttal. All of my questions have been answered with good answers and extensive qualitative results, and I am happy to raise my score.

---

### Comment · Area_Chair_Y9jq · 2025-08-04
**Please carefully read the rebuttal and start the discussion**

Dear Reviewers and Authors,

Thank you all for your efforts so far. As the author–reviewer discussion period will conclude on **August 6**, please start the discussion as soon as possible.


**For Reviewers:**
Please read the authors’ responses and, if necessary, continue the discussion with them.

* If your concerns have been addressed, consider updating your review and score accordingly.

* If some concerns remain, or if you share concerns raised by other reviewers, clearly state these in your review and consider adjusting your review (positively or negatively).

* If you feel that your concerns have not been addressed, you may also choose to keep your review as is.

* I will follow up with you again during the reviewer–AC discussion period (August 7–13) to finalize the reviews and scores.


**For Authors:**
If you have not already done so, please respond to all questions raised by the reviewers. Keep your responses factual, concise, and ensure that every point raised is addressed.

Best regards,

The AC

---

### Decision · Program_Chairs · 2025-09-17

**Decision:**

Accept (poster)

**Comment:**

This paper investigates memorization in multi-modal (image–text) foundation models using a controllable synthetic dataset. Carefully designed experiments test whether models can recognize facts in both modalities when exposed to them in only one modality during training. The results show that knowledge transfer across modalities is incomplete: neither larger model sizes nor diverse training variations close the gap, though additional in-distribution image–caption pairs improve performance. The study also reveals asymmetries depending on which modality serves as the source.

All reviewers found the work interesting and the findings significant. Major concerns were resolved during the rebuttal, and the authors are encouraged to incorporate all feedback in the camera-ready version.